# Defining “Normal” in Pig Parturition

**DOI:** 10.3390/ani12202754

**Published:** 2022-10-13

**Authors:** Alexandra Walls, Bianca Hatze, Sabrina Lomax, Roslyn Bathgate

**Affiliations:** 1Sydney School of Veterinary Science, Faculty of Science, The University of Sydney, Sydney 2006, Australia; 2School of Life and Environmental Sciences, Faculty of Science, The University of Sydney, Sydney 2006, Australia

**Keywords:** behaviour, eutocia, dystocia, reproduction, sow, welfare

## Abstract

**Simple Summary:**

The dual considerations of efficient food production and good animal welfare have never been so important nor under such strong public scrutiny as they are in current times. Intensive animal production industries play an important role in feeding an ever-growing, increasingly affluent population that is hungry for animal proteins. The efforts to improve efficiency in these production systems must not come at the cost of animal welfare. One pain point in pig production is that of parturition, where factors such as litter size and sow-housing type have been shown to influence the process and outcome. However, there are still many gaps in knowledge surrounding the normal physiology and endocrinology in a farrowing and this review seeks to summarise what is known and highlight areas where further work is required.

**Abstract:**

Animal production industries rely on efficient and successful reproductive outcomes, with pigs being no exception. The process of parturition in pigs (farrowing) can be especially prolonged, due to the large numbers of piglets being born (on average, approximately 13 piglets per litter in Australian conditions). Difficulties in farrowing (dystocia) lead to poor piglet outcomes and health problems in sows, in turn, causing economic loss for producers and welfare concerns for the animals. Despite the importance of this topic and publications in the area stretching back nearly 50 years, there is still no consensus on the prevalence of dystocia in pigs nor on how to identify a pig experiencing the condition. Understanding the process of parturition and the factors that influence its success is a crucial step towards the early identification of sows undergoing dystocia and development of best practices to assist them. This article describes the key factors that contribute to successful farrowing and identifies areas in which more research is required before the parturition process in the pig can be fully understood.

## 1. Introduction

An understanding of mammalian parturition is important for animal production industries as it is one of the keystones for success in this setting, which relies on efficient reproductive outcomes. Although much research has focused on the prevention of problems during parturition (dystocia) in ruminants (reviewed by [1,2]), relatively little attention has been given to the pig in this area. This is perhaps due to the common perception that the frequency of dystocia in this species is low. Historically, rates of less than 5% have been reported [3], but the definition of normal parturition in the pig has not been identified adequately and reported ranges of “normal” are widely disparate.

The limited reports on parturition in the direct ancestors of the domestic pig, the wild pig, suggest that the average litter size is about 6 piglets [4,5] and both the farrowing length and inter-piglet interval vary greatly, but the averages are about 81 min and 26 min, respectively [5]. Additionally, data from studies in domestic pig parturition from several decades ago can no longer be relied upon, as they were essentially studying a different animal due to the vast increase in litter sizes and neonatal growth rates resulting from the genetic selection that led to the development of the modern pig. These modern pigs are defined by characteristics such as increased growth rates, a leaner carcass, larger litter sizes, and more stillborns than their counterparts from several decades ago [6,7,8].

It is clear that periparturient events influence piglet survival [9]. It is also well documented that difficulties in parturition lead to compromised sow health, with an increased chance of developing postpartum dysgalactia syndrome (PPDS) observed with the increased duration of parturition and subsequent decreased chance of successful rebreeding [10]. This, along with the aim of improving the welfare of the sow in this period, demonstrates the need to develop an understanding of eutocia in the pig. This review aims to outline the current understanding of factors that influence parturient success in the pig while highlighting areas in which there is a knowledge gap requiring further study.

## 2. The Process of Parturition

Mammalian parturition has been studied most intensively in the human, but the main concepts can be transferred to most other mammals. The two key events required for a successful parturition are the opening of the birth canal via softening and dilation of the cervix (cervical ripening) and the removal of the myometrial blocks to allow for uterine contractions [11]. Parturition is composed of three stages, the first being the preparatory stage of cervical ripening and the onset of myometrial contractions to position the first fetus in the apex of the birth canal. The second stage is defined by the appearance of strong abdominal contractions, rupture of the allantochorionic sac and expulsion of the fetuses. The third and final stage includes the expulsion of any remaining fetal membranes (these are sometimes expelled alongside the fetus in polytocous species) [12].

It is important to note the significant pain associated with this process [12]. Labour pain is ranked as one of the most intense pains recorded in humans [13]. Based on their similar anatomy, central nervous system and responses to noxious stimuli, it is reasonable to assert that other mammals experience pain during parturition [14]. Following this, it could also be inferred that as in humans, dystocic parturition in pigs causes a significant amount of pain [12]. This concept is supported by studies showing that sows experience pain in parturition [15] and is worse in cases of difficult farrowings [14]. Definitions of normal parturition (eutocia) and dystocia should be established to quantify the impact on reproductive outcomes on the farm and allow for appropriate intervention such as pain relief to sows. This will improve welfare and enable better piglet outcomes.

### 2.1. Pregnancy in the Pig

The length of gestation in the pig is renowned for being “3 months, 3 weeks and 3 days”, but the actual time frame varies between breeds and individuals to be between 105 and 125 days [16]. The fetuses are evenly distributed throughout the bicornate uterus [17], and placentation is classified as anatomically diffuse and histologically epitheliochorial [18]. The corpora lutea (CL) are present throughout the duration of the pregnancy and remain the primary source of progesterone throughout this time [19]. The average litter size varies between breeds, strains and husbandry systems, with examples being 11 in the USA [20] to nearly 15 in Denmark [21].

### 2.2. Eutocia and Dystocia

Eutocia in the pig can be defined as the successful expulsion of all fetuses and placentas without the aid of obstetrical intervention. Dystocia describes a parturient experience that is slow or difficult, often requiring assistance. Current working definitions of dystocia in the sow rely heavily on a handful of accounts (summarised in Table 1). Even within the few publications on this topic, classifications are highly variable, with some suggesting a focus on inter-piglet intervals [22,23,24,25], whereas others suggest a total farrowing duration [26,27] or a combination of these and other factors. Notably, the figure of a 45–60 min inter-piglet interval is common across several publications (Table 1), but it is unclear from where this figure is derived. Equally contentious are prevalence rates, which range from 0.25–47% of spontaneous farrowing [22,27].

Despite the lack of consensus on prevalence, it is agreed that dystocia most commonly occurs in the presence of fetal obstruction or inadequate myometrial contraction activity [3,23]. Obstruction of the fetus within the birthing canal (reported to be responsible for 63% of dystocia cases) [28] can be caused by constipation, mispositioning of the fetus or maternal pelvis dimensions and, in some cases, a combination of these factors [28]. Inadequate myometrial contractions at farrowing are reported to be the cause of the remaining cases of dystocia (37% of cases) [28] and may be caused by fatigue, uterine inertia, hormonal imbalances and nutritional irregularities [3,23]. With the steady increase in litter size and associated physiological demands for the sow in modern production systems [6], additional research is warranted for timely identification and treatment administration to animals in need of assistance. Notably, further understanding of myometrial activity, appropriate inter-piglet intervals and behavioural dystocia identifiers are necessary to improve both sow and piglet welfare [29].

**Table 1 animals-12-02754-t001:** Proposed definitions and cited prevalence of dystocia in the pig by previous authors.

Reference	Dystocia Classification	Prevalence (%)
Nam and Sukon 2021 [30]	Inter-piglet interval greater than 45 min, application of obstetric assistance	47
Zamemba et al. 2019 [25]	Inter-piglet interval greater than 45 min	11.5
Oliviero et al. 2010 [26]	A farrowing greater than 300 min	N/A
Cowart 2007 [23]	Failure to deliver fetuses within 2 h from the onset of labour, an inter-piglet interval greater than 1 h, a gestation period beyond 116 days, sow illness, discoloured vulval discharge	1
Alonso-Spilsbury et al. 2004 [24]	Absence of uterine contractions, inter-piglet intervals greater than 1 h, application of obstetric assistance	N/A
Jones 1966 [27]	Labour lasting longer than 2 h in duration, application of obstetric assistance, uterine inertia	0.25

## 3. Endocrinology of Parturition

The details of the key factors that contribute to a successful farrowing in a normal birth have not been defined in the modern sow. Creating reference ranges for these is important to differentiate between eutocia and dystocia and will enable an understanding of which are most significant for piglet survival and sow welfare. There are also external factors that contribute to the experience of parturition in the pigs, such as the type of housing and diet. These will also be discussed.

### 3.1. Periparturient Endocrinology

The endocrine cascade that initiates and controls parturition differs between mammalian species, and in several species, begins with the increased secretion of cortisol from the fetus. In the pig, an increasing concentration of adrenocorticotropic hormone (ACTH) from the fetal anterior pituitary (adenohypophysis) acts via the fetal hypothalamic–pituitary–adrenal (HPA) axis to cause secretion of glucocorticoids from the adrenal cortex. This likely has a significant role in fetal maturation [31], but does not seem to be the ultimate trigger of parturition in the pig. The increased cortisol redirects endometrium-derived PGF2α from the uterine lumen, where it is sent under the influence of embryonic oestradiol during pregnancy to the uterine vein and, subsequently, into general circulation [32]. This access to the uterine veins gives direct transmission to the ovarian arteries and, thus, initiation of luteolysis. The dominance of luteal-derived progesterone on the maternal reproductive tract is lost with the onset of luteolysis [31]. Alongside this, the conversion of any progesterone remaining in the system to oestrogen by the placental-derived enzymes sees a dramatic switch in the ratio of progesterone: oestrogen. This, in turn, removes the myometrial block that was imposed by progesterone and initiates contractions under the influence of oestrogen. These first contractions push the first fetus down onto the internal os of the cervix and this stimulation creates a positive feedback loop with release of oxytocin from the posterior lobe of the pituitary (neurohypophysis). Oxytocin stimulates powerful myometrial contractions, forcing more pressure onto the cervix from the fetus and, thus, stimulating more oxytocin to be released from the pituitary [33]. Simultaneously, the increased oestrogen in the system increases mucosal secretion in the reproductive tract, which creates the lubrication needed for passage of the fetuses. Finally, relaxin secretion from the CL peaks in the days just prior to parturition, enabling enhanced stretching of the pelvic ligaments to accommodate the passage of the fetus through the birth canal [34]. The pattern of changing concentrations of these key hormones is summarised in Figure 1. However, these data should be viewed with the erudite comment of Gilbert, 2001 [35], in mind; changes of hormone concentration in the circulating blood are not fully informative without also knowing the relevant receptor expression. This is where there continues to be a scarcity of data, preventing full understanding of the control of parturition in the pig.

#### 3.1.1. Oestrogen and Progesterone

Parturition cannot proceed without the removal of the myometrial block imposed by high concentrations of circulating progesterone during pregnancy [36]. In the pig, luteolysis (the CL being the main source of progesterone throughout gestation [37]) occurs about 24 h prior to parturition. With an intravenous half-life of approximately 30 min [38], the circulating progesterone concentration rapidly declines when it is no longer being replenished. The corticosteroid parturient trigger stimulates production of oestrogen by the placenta [39]. This, alongside the cessation of progesterone production due to luteolysis and the conversion of progesterone to oestrogen by the placentally derived enzymes [40] over the days prior to parturition, are the drivers behind the dramatic transient rise in oestrogen that initiates uterine contractions in stage 1 of parturition. These preliminary contractions assist in positioning the first fetus at the internal os of the cervix, applying pressure that leads to the release of oxytocin from the posterior lobe of the pituitary gland. Oestrogen also stimulates the formation of oxytocin receptors in the uterus [41] that work to drive myometrial contractions during farrowing.

#### 3.1.2. Oxytocin

Oxytocin (Ot) is a neuropeptide produced predominately in the hypothalamus and stored in the posterior lobe of the pituitary gland, but Ot synthesis also occurs in the decidua [42]. The effects of oxytocin and its mechanisms of action are still being elucidated despite being extensively studied in humans, making the interpretation of data on Ot concentrations and the presence of receptors problematic (reviewed by [43]). However, throughout human pregnancy, the release of Ot is slow and circulating levels are kept low by the action of oxytocinase, an enzyme produced by the fetal membranes and placenta. Additionally, myometrial receptivity to oxytocin appears to remain low until the latter stages of pregnancy [41]. In the pig, oxytocin concentrations begin to rise about 7 h before the birth of the first piglet (reviewed by [44]). Increased E2 concentration and pressure on the cervix at the time of parturition induce the pituitary to release large amounts of Ot in a pulsatile manner, as well as increasing the synthesis of Ot from the decidua. Additionally, the increased blood concentration of oestradiol likely further stimulates myometrial contractions. This acts to drive the intense contractions associated with stage 2 of parturition via an autocrine pathway, stimulating production of the prostaglandins that then also act directly on the myometrium to cause contractions. Additionally, an increased expression of Ot receptors is seen on the myometrium in response to the switch in the progesterone: oestrogen ratio and to uterine stretch [45]. It should be noted that probably due to redundancy, Ot does not seem to be vital for successful parturition, but clearly has a key role within normal systems [46]. Tangentially, Ot appears to also have a role in the protection of the fetus from hypoxia during parturition [47].

#### 3.1.3. Prostaglandin F2α and Prostaglandin E2

These two prostaglandins (PGs) appear to be the main members of the large prostaglandin family that influence parturition. In the pig, where CL are present throughout gestation, a rise in circulating PF2α, probably derived from the placenta, causes both the expression of nesting behaviour via the increased secretion of prolactin and luteolysis in the days immediately preceding parturition [48]. This not only releases a reservoir of relaxin (see Section 3.1.4) stored in granules within the luteal cells [49], but also leads to a fall in progesterone concentrations (see Section 3.1.1) as the main source of this hormone being removed. The other direct effect of PGF2α is the contraction of myometrial cells. In conjunction with oestrogen, the resulting pressure on the cervix as the fetuses are pushed through the uterus leads to the excretion of oxytocin from the posterior lobe of the pituitary (see Section 3.1.2). Prostaglandin E2 is known to stimulate the maturation of the fetus in preparation for expulsion in humans [50] and probably pigs. It also has a role in cervical ripening and increasing the intensity of myometrial contractions during parturition [51]. Importantly, both may also have a role in the pain of parturition. Administration of PGF2α leads to acute inflammation via augmentation of arterial dilatation and increased microvascular permeability, whereas PGE2 sensitises peripheral sensory neurons and central sites within the spinal cord and the brain [52].

#### 3.1.4. Relaxin

The corpora lutea are stimulated to release a surge of relaxin around 12 h prior to the onset of parturition [53]. This is stimulated by the preceding increase in PGF2α [54]. The increase in relaxin concentration appears to extend the time of uterine myometrial quiescence past the observed fall in progesterone, as myometrial contractions do not commence until both progesterone and relaxin levels fall [55]. This delay in myometrial activity appears to be the main role of relaxin in the pig, although it also plays a role in inducing cervical softening and dilatation [56] and coordinating uterine contractions during farrowing [55].

#### 3.1.5. Prolactin

The concentration of prolactin in the blood increases gradually prior to parturition, to eventually peak at the birth of the first piglet [33]. This hormone, secreted from the anterior pituitary, has been linked with stimulation of nesting behaviour before parturition by some authors [57], but not others [58]. It does have a role in stimulating mammary development in preparation for lactation [59] and milk production postpartum [60].

## 4. Factors Affecting Parturition

### 4.1. Uterine Acticity and Contractility

For success at parturition, smooth muscle associated with both myometrial and abdominal contractions must work consistently throughout parturition [61]. In the sow, the understating of myometrial activity throughout the different stages of parturition is limited and relies heavily on a handful of accounts [12,62,63,64,65]. Electromyography (EMG) of the uterus before and during parturition indicates three main uterine changes, each occurring rapidly after one other [12,63]. The first, described as the pre-parturient phase, is a shift in normal myometrial activity 4–9 h before the first fetal expulsion [63]. Contractions at this stage regularly persist at a steady amplitude and duration [63]. Myometrial frequency and intensity increase within the last couple of hours in preparation for the birth of the first piglet and are accompanied by straining from the sow [63]. The second phase of parturition signifies an increase in mean contraction intensity and amplitude as the body expels fetuses as promptly as possible [62]. The third, described as the post-parturient phase, includes placental birth where contractions persist frequently but at a decreased amplitude [12,63].

Disruption to activity, commonly called uterine inertia, can cause significant risk to both sow and piglets. Uterine tone is a relatively unexplored factor potentially contributing to the duration and experience of parturition between parities. In women, parity and associated muscle tone of the reproductive tract correlate with parturition success [66]. When examined using EMG, nulliparous women had higher contractility of the reproductive tract than multiparous women [66]. In the sow, little is known about the effect of muscle tone and its association with dystocia occurrence. There is better understanding of this in the cow, where it is believed hypocalcaemia, hormonal disturbances, infection and genetic weakness of the smooth muscle play a large part in reproductive muscle contraction at parturition [61]. Improved understanding of the role that muscle tone plays on success in parturition of the sow is important for improving protocols for management of farrowing within the industry.

### 4.2. Duration of Parturition

The duration of parturition in the sow is one of the factors that varies most markedly between breeds, individuals and parities. Total parturition time will be strongly dependent on the litter size, but recommendations of what should be defined as normal for total time and time per piglet are important due to the direct implications to piglet health and survival and sow wellbeing.

#### 4.2.1. Overall Parturition

Genetic selection for larger litter sizes has seen parturition duration rise over the past five decades. Research undertaken 50 years ago describes the experience to last, on average, 2.53 h, but notes a significant degree of variability (ranging from 56 min to 8 h 55 min) between individual sows [27]. More recently, the average parturition time of the modern hyperprolific sow has been described as an average of 3.58 h [67]. Between 1966 and 2019, average litter sizes have increased from about 11.5 to 14 piglets [27,68]. A change in the duration of parturition is expected as sows expel a greater number of piglets. With the demands that a larger number of fetuses place on physiology, it is no surprise that estimations of dystocia rates have risen from 0.25% to 5% in the modern hyperprolific lines of today’s production system [68]. In sows experiencing longer farrowing durations, abnormal piglet positioning, uterine inertia and obstruction of the vaginal canal are often observed [6]. Langendijk [69] noted a significant increase in piglet mortality when farrowing duration exceeds 8 h, a 24.6% increase over sows farrowing in 2 h or less. Additionally, in those that survive prolonged farrowing, extended durations have subsequent effects on sow health and welfare, reduced appetite, fever, reduced colostrum letdown, and increased sow removal (i.e., wastage) [68,70]. Poor farrowing can increase the risk of postpartum dysgalactia syndrome (PPDS), a severe and common illness that presents in the sow shortly after farrowing, which is associated with an interrupted farrowing, reduced weaning weights and increased piglet mortality [71]. Reproductive failure due to dystocia-associated illness such as PPDS is believed to contribute to high turnover rates (as high as 61% in Australian conditions) [72]. Several cross-farm producer questionnaire studies found that between 44% and 67% of sow cullings were performed on animals with a parity of three or less [73,74,75,76]. The early culling of sows was attributed to large stillborn rates, undesirable litter sizes and disease [73]. With sows reaching their peak production between the third and sixth parities, the undesirable culling of young production animals is costly to the production chain [73]. Therefore, it is imperative for the success of the porcine industry and the welfare of both sow and piglets that the parturient process for the modern sow is thoroughly understood. Particularly, a reasonable farrowing timeframe requires establishment within the modern industry to permit treatment administration to those farrowing outside appropriate windows.

#### 4.2.2. Inter-Piglet Interval

Inter-piglet interval is defined as the time between the expulsion of one piglet and the next [77]. It is generally accepted that most sows will farrow with piglet intervals of 15–22 min [27,29,78,79]. Piglets born in the latter half of farrowing are observed to experience longer interval times than those born in the first half of a litter [27,80]. This could be due to sow exhaustion or reduced contraction intensity as farrowing progresses. Piglets experiencing a longer birthing interval have an increased risk of morbidities due to hypoxia leading to death in severe cases [79]. Correspondingly, sows who present with extended straining without the appearance of a piglet are prone to exhaustion, uterine infection and potentially death if appropriate intervention does not occur [6].

There is some dissent around when it is appropriate to intervene in cases of an exaggerated inter-piglet interval, but it is evident that without treatment, the risk to both sow and piglet is high. Currently, the advice is to manually intervene if 45 min passes after the birth of a piglet without the subsequent appearance of the next [6]. However, variation between individual sows makes defining a gold standard for intervention difficult. Although there is a notable reduction in stillbirths when manual intervention is undertaken, long-term effects of potentially unwarranted and often nonsterile techniques are questionable [81]. Further research is required to either validate or improve existing intervention protocols for enhanced sow and piglet welfare.

### 4.3. Farrowing Accommodation

Housing type influences the process and outcome of farrowing [82,83,84,85]. In the field of human obstetrics, it has been shown that labour proceeds smoothly and rates of dystocia are reduced when women are encouraged to remain mobile throughout the first stage [86]. However, most sows in intensive production are confined in housing that restricts movement to reduce piglet fatalities and morbidities by savaging and overlay [83]. The trade-off between sow welfare and piglet safety is a complex one, and sow comfort during parturition should be one of the factors considered.

#### 4.3.1. Freedom of Movement

The farrowing crate commonly used in modern, intensive production systems limits sow movement at farrowing, allowing piglets time to escape crushing when a sow lies down [83]. Although successfully reducing overlays, this housing type has been shown to prolong farrowing duration and increase stillbirth numbers [84], as well as reducing sow welfare, as measured by circulating cortisol concentration [87]. In contrast, the farrowing pen provides sows with the ability to move freely, but gives a greater opportunity to injure piglets [83]. A reduced farrowing duration and inter-piglet interval in sows housed in pens compared to crated systems has been observed, suggesting a less restricted environment at parturition has positive effects on progression [26,82,83]. This has a direct impact on the experience and success of farrowing, as discussed in Section 4.2.

One aspect of research in this area has been in the modification of the housing used in farrowing to minimise sow confinement without an increase in piglet mortality by modifying the shape of crates to incorporate anti-crush attachments. Temporary crating systems (sometimes called free-farrowing systems) allow for sows to be confined for a shorter duration when the chance of piglet overlay is greater, before the system is converted into a pen configuration to give greater freedom of movement [88]. When comparing free-housed sows to sows housed in one of these systems (Sow Welfare and Piglet Protection; SWAP pens), behavioural differences were minimal, suggesting short-term confinement could be the happy medium between piglet and sow welfare at parturition [88,89]. There are moves in some countries towards legislating the use of these pens [90], driven in part by the “End of the cage age” Citizens’ Initiative in the European Union [90]. Increased scrutiny of welfare in the pig industry means temporary crating systems could be implemented globally in the future.

#### 4.3.2. Availability of Nesting Material

The benefits of providing nesting material to sows at farrowing has long been demonstrated [91]. Like their wild relatives, many domestic sows instinctively perform nest-building behaviour, including pawing, rooting and seeking nesting material in preparation for piglet arrival [92]. Such behaviours are believed to support endogenous hormonal changes at partition, particularly the secretion of oxytocin [84]. When these behaviours are hindered, the endocrine profile is altered, causing increased stress and, subsequently, reduced welfare. As Oliviero et al. [54] reported, sows unable to express natural nest-making behaviours experience increased circulating cortisol (a biomarker of stress) prior to farrowing and reduced oxytocin secretion during the expulsive period of parturition. Additionally, Oliviero et al. [84] noted an average 1.5 h increase in the duration of parturition and an extended inter-piglet interval in sows unable to perform nesting behaviours compared to those in free housing with access to nesting materials. Plush et al. [93] observed a reduced incidence of piglet mortality when sows housed in conventional farrowing crates were provided with foraging materials over those without. Where loose nesting material is not feasible due to drainage systems, larger materials less susceptible to falling into grates, such as hessian sacks, may be used. In order to improve both sow and piglet welfare, it is important to enable the farrowing sow to exhibit some natural behaviours.

#### 4.3.3. Environmental Conditions

For optimal farrowing, the local environment should be free from external stressors. Increased plasma cortisol concentrations in the sow when ambient temperatures rise above 25 °C warrants careful management within production systems [94]. The evaporative critical temperature for the sow is 22 °C, and temperatures above this stimulate evaporative cooling through increased respiration from the lungs [95]. Stress associated with high ambient temperatures prolongs the farrowing duration and reduces appetite at lactation [95,96]. Elevated relative humidity levels exasperate the effect of high air temperature, making adaptive cooling difficult for the sow [97]. Hence, the temperature within farrowing sheds should be monitored closely and maintained below 22 °C with a relative air humidity between 60–70% to minimise stress and energy usage [95,97].

Moving sows and gilts to farrowing units between days 95–105 of gestation allows for their habitation to new surroundings and significantly reduces restless behaviour before parturition compared to the movement of sows at day 114 of gestation [98]. The introduction to farrowing units too close to parturition can alter maternal behaviour and disrupt the farrowing process, risking the welfare of both sow and piglets [98]. In systems where temporary crating exists, additional care can minimise stress, particularly in primiparous animals experiencing the parturient process for the first time. Crating gilts and sows after parturition has commenced increases farrowing duration [99]. In contrast, confining animals two days before farrowing inside their loose-housed pens does not affect the duration of parturition [89].

### 4.4. Sow Parity

There is some speculation surrounding the effects of sow parity, particularly the differences between multiparous animals (parities 2+), on the parturient process of the modern sow. Putting gilts (parity 1) aside, several authors provide evidence to suggest there is no significant interaction between the parity of a sow and the duration of farrowing [26,29,100]. Contrastingly, others contest this and present evidence to support farrowing duration increasing alongside sow parity [101,102,103,104]. Ju et al. [102] suggest this could be due to the aging of the uterine muscle tissue after multiple farrowings, reducing the ability to contract, and hence, prolonging the parturient process. It was also shown that factors such as litter size, body condition and environment are highly correlated with parturition success and determining the effect of parity alone is difficult [102]. Although the effect of parity is under debate, it is agreed that the parturient experience is highly variable between animals and, as such, each case should be observed objectively to ensure both sow and piglet welfare.

Contrastingly, the gilt (parity 1) is perceived as a separate consideration when examining potential farrowing success. During parturient periods, gilts often present with enhanced agitation (continuous standing up and lying back down) over higher parity sows [105]. As a result of increased stress, savaging events are more likely and tend to occur within the birth of the first two piglets [105]. Early savaging in gilts has been attested to their lack of familiarity with the birthing process and has been tied to farrowing in a confined environment [106]. Although the progressive portion of parturition often develops more promptly in gilts than sows of higher parities, the total duration is similar irrespective of parity [105]. An early study suggests that stress experienced by gilts at first farrowing directly correlates with oxytocin secretion, which would otherwise be opioid-mediated, subsequently enhancing the speed at which the pre-parturient phase progresses [107]. To ensure success at parturition in the gilt, efforts must be placed on monitoring parturition progression and reducing any unnecessary additional stressors.

### 4.5. Sow Nutrition

The dietary intake of a sow influences the progression of farrowing [26,108,109,110,111]. Vigilant monitoring of the sow body condition during gestation is necessary to ensure correct back-fat coverage at farrowing [26]. Sows with body condition scores above 4 (more than 21 mm back fat) are often reluctant to move during farrowing. Reduced movement at parturition is linked to increased farrowing times and is speculated to be due to the inability to contort fetuses within the birthing track [83]. Conversely, sows with too little back-fat coverage (<16 mm) risk inadequate energy reserves for high-energy bodily processes such as parturition and lactation [108]. Consequently, sows must be managed appropriately, ensuring even fat coverage throughout gestation for success at farrowing.

Associated with sow diet, acute constipation increases pain and discomfort at farrowing, leading to extended farrowing times [26]. Constipation within the sow is common during periods leading up to and during farrowing as the digestive system reduces activity in preparation for fetal expulsion [109]. However, extensive periods of abnormal defecation can indicate further issues such as inadequate water intake [109]. This is especially important as the sow requires a higher than usual intake of water at this time in preparation for milk production [112]. Increased solids within the rectal tract can increase the risk of fetal obstruction, as they create a force against the adjacent birthing canal [109]. Additionally, prolonged constipation increases the absorption and release of bacterial endotoxins, which have been linked to postpartum dysgalactia syndrome [110]. To mitigate risks of constipation at farrowing, sows should be offered feed with increased crude fibre (7–10%) during late pregnancy [111]. Well-informed decision-making and daily observations of digestive inconsistencies can improve the parturient experience for the sow, increasing the likelihood of success at farrowing.

### 4.6. Piglet Factors

Alongside the impact of maternal physiological processes, there is a general consensus that piglet factors such as size, presentation and number of fetuses can also significantly impact parturition duration and success.

#### 4.6.1. Piglet Size

Larger piglets take, on average, longer to be expelled than smaller ones [81,113,114]. This is thought to be a result of increased friction between the fetus and the birthing canal as it move through the pelvic area [114]. However, Rens and Lende [105] found placental thickness to incur a greater effect on enlarged birthing intervals than the weight of individual piglets. Smaller piglets in litters with large size variation have also been observed to have a higher stillbirth probability [81,113,114,115]. This may be due to the increased rate of abnormal birthing positions at farrowing in piglets of a smaller size, or perhaps, reflects a disadvantage due to resource competition with their larger litter mates in utero. Additional space around small piglets allows easy contortion of limbs and body parts as cervical contractions push them through the birthing canal. An even spread of piglet size within a litter may reduce the occurrence of prolonged farrowings [81]. Although some effort has been undertaken to determine the effect of piglet size on the parturient experience, its influence on the incidence of dystocia remains unresolved.

#### 4.6.2. Piglet Presentation

Piglet presentation within the birthing canal can affect the parturient experience for sows and piglets. Presentation can be classed in five ways (Figure 2): (1) The cranial presentation of piglets involves the forelegs gathered up against the body, with the head exiting first. Full extension of the fetal body and limbs allows for free movement along the birthing canal, making cranial presentation the most desirable of fetal positions. (2) The caudal presentation of piglets appears as hindlegs first, posteriorly extending away from the body. Piglets born caudally experience raised lactate levels, pH values, exasperated inter-piglet intervals, and increased quantities of epinephrine within umbilical cord blood compared with piglets born cranially [29,116,117]. This indicates increased physiological stress, an undesirable addition to an already challenging experience that is parturition [29]. (3) A breech presentation involves the hindquarters exciting first, with hindlimbs flexed forward towards the crown of the fetus. (4) A transverse presentation consists of a lateral positioning of the body with fore- and hindlimbs exiting last. (5) A poll presentation is characterised as a bent neck with the head pressed against the body and snout facing towards the cervix.

Most piglets born are delivered in a cranial position (approximately 50%), with caudal presentation reported the second most frequently [116]. Observed less often within pig parturition, breech, transverse and poll presentations can incur additional challenges at fetal expulsion. Breech, transverse and poll presentations of the fetus increase chances of lodgement within the birthing canal, extending inter-piglet intervals [118]. Considerable variation within fetal piglet sizes may increase the occurrence of undesirable fetal presentations [81,117]. Ensuring adequate nutrition during parturition for fetal development and selection for sows with reduced litter variability may lessen the occurrence of issues associated with undesirable piglet presentation.

#### 4.6.3. Litter Size

The selection for increased litter size across the industry has added additional strain to sow reproductive anatomy and physiology [119]. In a little over five decades, litter size has grown by an average of 2.5 piglets from 1966 to 2019, with a litter size up to 22 piglets reported [6,27,120]. Larger litters take a greater length of time to farrow than smaller litters, directly affecting the occurrence of piglet asphyxiation and sow dystocia. Expelling a larger number of fetuses can increase the risk of uterine inertia, lodgement of fetuses and misconfiguration of piglets in the reproductive tract [6]. Additionally, infection or injury of the reproductive tract due to increased physiological demands of enlarged litter sizes can affect future sow reproductive performance [66]. To counteract risks associated with larger litters, monitoring sow wellbeing, inter-piglet intervals and piglet activity levels at farrowing is advisable [78,79]. Prompt detection and implementation of aid to animals experiencing prolonged farrowings can improve the welfare of both sows and piglets undergoing parturition under increased production pressures.

## 5. Induction of Parturition

The preceding information describes the situation in a naturally induced parturition. It is important to note that in many production systems, it is common to induce farrowing either for ease of management or veterinary intervention when sows are experiencing an extended pregnancy. For example, a survey of UK-based pig producers in 2016 demonstrated that approximately 15% of respondents often induced parturition [121]. Farrowing induction is commonly performed by the administration of PGF2α or an analogue around the anticipated due date to initiate luteolysis and, therefore, trigger parturition. However, the regime of administration varies, with single, double or multiple doses at varying concentrations [122]. In some instances, this PGF2α is supplemented with the subsequent administration of oxytocin or an analogue to stimulate myometrial contractions [123].

The most common reason for induction is because it is desirable to have sows farrow in standard working hours, to enable more cost-effective supervision. The increased observation of farrowings improves piglet outcome by reducing stillborn and postnatal mortality rates [78,124]. This improved outcome from the supervision of farrowing can be offset by a lower piglet survival rate when sows are induced [78], especially when induction occurs too early in gestation [122,123], and by increased rates of PPDS in sows [125]. However, a recent meta-analysis has concluded that under the right conditions, the induction of parturition in sows with PGF2α or an analogue has no effect on piglet outcome [122] and may be beneficial in allowing for the greater observation of sows when parturition occurs in daylight hours.

## 6. Concluding Remarks

Mammalian parturition is a complex process with many factors influencing the outcomes, especially in polytocous species such as the pig. Key concepts to consider when defining “normal” parturition in pigs include parity, litter size, piglet presentation and farrowing accommodation. These will influence factors such as duration of parturition, inter-piglet interval and, therefore, overall piglet survival. Other factors not discussed here that may also be important include overall sow fitness and, perhaps, breed differences. It is clear that there are gaps in our understanding of the endocrinology and physiology of this process that should be filled to optimise piglet survival and sow welfare. These gaps prevent the development of a working definition of dystocia, but the definition will be a multidimensional one, including, at the least, inter-piglet interval and total duration of parturition, relative to litter size. Defining this will enable the creation of guidelines to allow timely and meaningful interventions to occur only when required.

## Figures and Tables

**Figure 1 animals-12-02754-f001:**
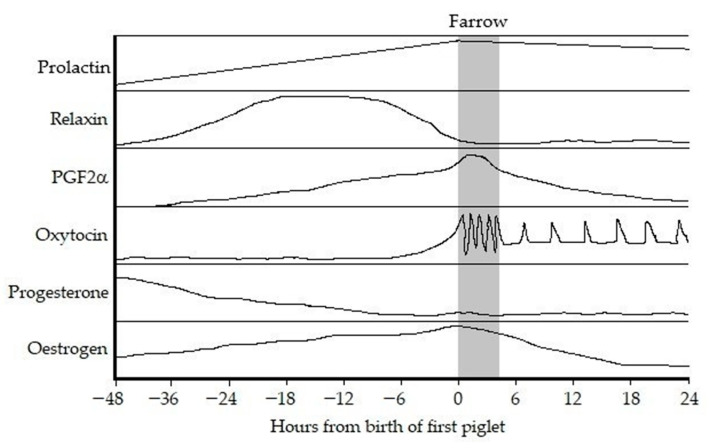
Patterns of periparturient plasma hormone concentrations in the pig. Unscaled. Adapted from [35] and used with permission.

**Figure 2 animals-12-02754-f002:**
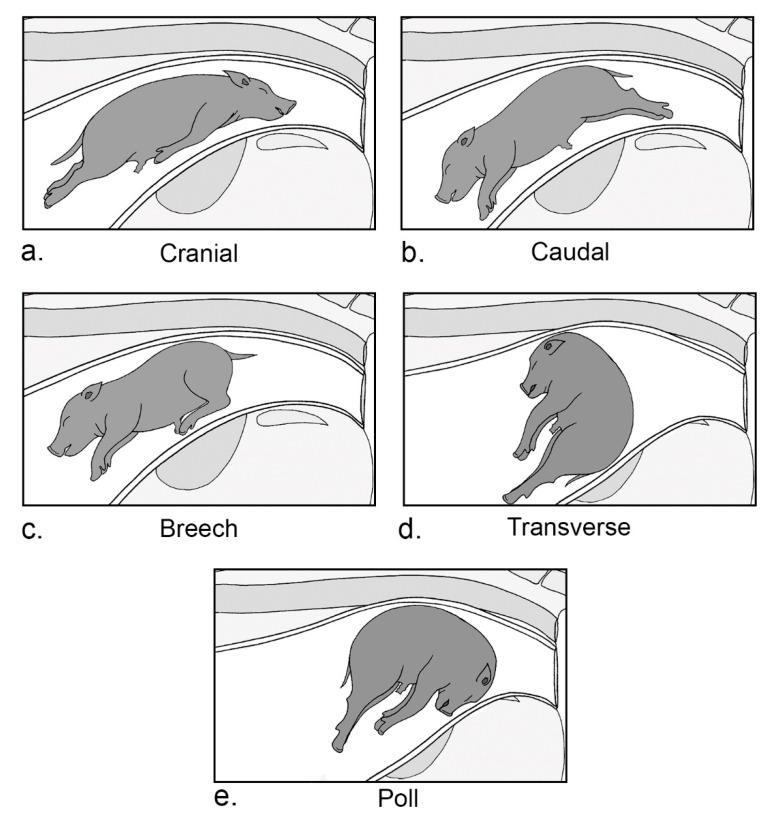
Anatomical piglet positioning within the birthing canal at parturition: (**a**) cranial presentation involving the head exiting first and forelimbs gathered up against the body; (**b**) caudal presentation involving hindlegs first, posteriorly extended away from the body; (**c**) breech position involving the hindquarters exciting first and hindlegs flexed forward towards the body; (**d**) transverse positioning involving a sideways positioning of the body with fore- and hindlimbs exciting last; (**e**) poll presentation involving a bent neck with the head pressed against the body and snout facing towards the cervix.

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
