# Peer review of "Defining “Normal” in Pig Parturition"

_animals, 2022, doi:10.3390/ani12202754_

Round 1

Reviewer 1 Report

Interesting title, encouraging to read, while the content is a bit disappointing because of the textbook approach. A very big advantage of the article is paying attention to farrowing accomodation, freedom of movement and the possibility of building a nest by the sow. I am glad that the authors pay attention to the issues of sow welfare, but I would suggest supplementing the information with the latest plans of the European Commission related to the implementation of "End of the cage age" (https://europa.eu/citizens-initiative/end-cage-age_en). This will make the publication more internationally valuable and as up-to-date as possible. Moreover, too little attention was paid to the role of prolactin, the relationship between farrowing crate (or farrowing pen) - prolactin level - duration of labor - sow care (it may be useful to add the changes in prolactin levels before farrowing, in the periparturient period and postpartum to the graph). It is positive that the authors drew attention to the relationship between the housing system (the ability to move the sow and build the nest) and the level of cortisol, but in my opinion, this thread should be expanded and enriched.

In a review-type publication, it is good to present the contrasting theories of different researchers in the form of a table. I would suggest adding such a table to present even more explicit topics that are worth exploring in the future to get clear, unambiguous answers. It would also diminish the impression of reading an academic textbook for students. In terms of content, it would also be worth referring to the course of delivery, perinatal behaviour in the ancestors of the domestic pig - wild boars, and to show the differences resulting from domestication and subsequent breeding / genetic improvement. Generally, I find this article enough valuable for publication after addition of mentioned above issues.

Author Response

Thank you for your comments. Please see below our responses as to how we addressed them.

Reviewer 1:

- I would suggest supplementing the information with the latest plans of the European Commission related to the implementation of "End of the cage age" (https://europa.eu/citizens-initiative/end-cage-age_en).

Information on this has been added to the “Freedom of movement” section.

- Moreover, too little attention was paid to the role of prolactin, the relationship between farrowing crate (or farrowing pen) - prolactin level - duration of labor - sow care (it may be useful to add the changes in prolactin levels before farrowing, in the periparturient period and postpartum to the graph)

The levels of prolactin in the periparturient period have been added to Figure 1 and a section (3.1.5) has been added to discuss the role of prolactin in this period.

 - It is positive that the authors drew attention to the relationship between the housing system (the ability to move the sow and build the nest) and the level of cortisol, but in my opinion, this thread should be expanded and enriched.

Within section 4.3.1, further information about the relationship between housing system and cortisol has been added.

- I would suggest adding such a table to present even more explicit topics that are worth exploring in the future to get clear, unambiguous answers

 A table has been added to highlight the lack of consensus in the current definition of dystocia

- it would also be worth referring to the course of delivery, perinatal behaviour in the ancestors of the domestic pig - wild boars, and to show the differences resulting from domestication and subsequent breeding / genetic improvement

A paragraph has been added in the introduction section (1) with information about parturition in the wild pig

Reviewer 2 Report

General comments.

An overview of the endocrine, behavioral and housing responses affect sows and piglets at parturition is presented. The primary responses are covered with a strong emphasis on the endocrine cascade that initiates parturition.  Mention and discussion of some common husbandry practices within the context of the focused area would strengthen interest in the review. For example, in discussions of oxytocin L131 to 136, What happens if exogenous interventions are applied, such as induction of timed parturition using PGF2a and oxytocin?

The entire manuscript needs to be edited to improve the academic style. Throughout the text phrases such as: “It is clear that”; “It is also well documented that”; It is important to note”; “It is reasonable”, “it” need to be deleted. Just simply make the statement with references cited to support the statement.

Specific Comments:

Line 103. Suggest…. dramatic switch in the ratio….

Line 104. Suggest …. progesterone and initiates contractions under….

L 141. Suggest …. being extensively studied in humans….

L 147. Define E2.

L 151. Has stage 1 been described? Stage 1 is mentioned in L 262.

 L 341-343. Not clear what you mean by dephosphorylation of ATP relative to energy reserves. Energy via oxidative metabolism produces ATP. If energy is depleted ADP (ie, dephosphorylation of ATP) accumulates. This section need to be revised for clarity.

L 341 to 345. As described readers may conclude that fat stores are the only sources of energy for muscle contractions. What about other energy substrates such as carbohydrates and excess amino acids that undergo oxidative metabolism to produce ATP.

Author Response

Thank you for your comments. Please see below a summary of how we have addressed them.

Reviewer 2:

- Mention and discussion of some common husbandry practices within the context of the focused area would strengthen interest in the review. For example, in discussions of oxytocin L131 to 136, What happens if exogenous interventions are applied, such as induction of timed parturition using PGF2a and oxytocin?

A new section has been added (6. Common interventions) with information on currently used protocols for intervention and the impact of these on the sow and piglets

  • The entire manuscript needs to be edited to improve the academic style. Throughout the text phrases such as: “It is clear that”; “It is also well documented that”; It is important to note”; “It is reasonable”, “it” need to be deleted. Just simply make the statement with references cited to support the statement.

The manuscript has been edited accordingly

  • Line 103. Suggest…. dramatic switch in the ratio…. corrected
  • Line 104. Suggest …. progesterone and initiates contractions under…. corrected
  • L 141. Suggest …. being extensively studied in humans…. Corrected
  • L 147. Define E2. Changed to oestradiol
  • L 151. Has stage 1 been described? Stage 1 is mentioned in L 262. This is described in section 2.
  • L 341-343. Not clear what you mean by dephosphorylation of ATP relative to energy reserves. Energy via oxidative metabolism produces ATP. If energy is depleted ADP (ie, dephosphorylation of ATP) accumulates. This section need to be revised for clarity. This section has been rewritten for clarity

  • L 341 to 345. As described readers may conclude that fat stores are the only sources of energy for muscle contractions. What about other energy substrates such as carbohydrates and excess amino acids that undergo oxidative metabolism to produce ATP.

This information has been removed in the rewritten section

Reviewer 3 Report

This is a comprehensive overview of the effects on pig parturition, although the authors have no experience in this area, based on the bibliography. In any case, a good work.

Suggestions:

line 19 - ...Australian conditions

line 27 - in keywords put pigs on the first place

Author Response

Thank you for your comments. Please see below a sumary as to how we have addressed them.

Reviewer 3:

line 19 - ...Australian conditions corrected

line 27 - in keywords put pigs on the first place As the word pig in used in the title, this seems superfluous

Reviewer 4 Report

The manuscript Animals-1927053, entitled "Defining normal in pig parturition" deals with a very interesting topic within the emerging field of animal welfare in sows. The manuscript is well structured, carefully formulated and presents interesting ideas. Various comments for changes to the text could be find in the following section.

General comment:

I would like to see a paragraph including the different definitions of “eutocia” and “dystocia” in pigs (at least up to now). And afterwards, based on your review, I would expect a more specific conclusion on this topic (further than you stated in L426-L428). Which is your proposal for eutocia/dystocia definition in pigs? Despite high variability between animals, is possible to propose which are the numbers to take under consideration at field level (in terms of duration of farrowing, frequency of manual intervention, piglet survival..)?

Specific comments:

L16: The abstract is relatively short. Maybe you can add some main results of the review to arouse the interest of the reader?

L32-L33: I would appreciate a reference supporting that sentence “much research has focused on prevention of problems in ruminants”

L36: Additional information about how “normal” parturition has been documented/defined up to now

L41: Are you referred to hyperprolific sows, or not necessarily?

L67: The mention of some papers based on surveys aiming to rank the different painful moments in pigs are welcome. For instance, Ison SH, Rutherford KMD 2014. http://dx.doi.org/10.1016/j.tvjl.2014.10.003 (and probably there are some more papers published more recently).

L99-L112: There is not any reference thought all these text. Please, add.

Section 2.1: A part of figure 1, maybe can help a table summarizing the 3 phases of farrowing, and explaining in each one: the duration, the main hormones, the main clinical effects (contractions, dilatation…)

Section 3.1.3: Taking into account that several farms are synchronizing farrowing using PGF2-alpha synthetic, it would be nice to mention it. And reviewing if that management can contribute to increase dystocia, or that depends on the day that is administered…or at least mention that there is another gap related to that (if that is the case). 

L182: (here or wherever) It would be nice to specify the different rhythm and intensity of the myometrial contraction depending on the moment of farrowing.

L231: It has been reported that the average parturition time is 3.58h for hyperprolific sows…and after how many hours is considered dystocia? After 5 hours for instance?

L217: Again, here you mention the dystocia rate, but it would be necessary to know how dystocia (or difficult farrowing) has been defined here.

L251: Good point, thank you!

L293: Not sure if I understand that sentence… “the secretion of oxytocin behaviours are hidered”..Please, check it.

New section 4.3.3: It would be nice to see a paragraph about environmental conditions that can affect farrowing. That should include the optimum temperature/humidity for farrowing, as well as the minimum number of days that sows can enter in the maternity unit in order to acclimate and adapt to the new building.

L336: What is considered high body condition? (introduce the mm of back fat coverage or at least the body condition score)

L352 (here or wherever): I suggest mention the importance of the administration of enough quantity of water around farrowing. Not sure if there is enough scientific information on this topic, but if not, it can be mention as an important gap to consider.

L378: Please, introduce reference of the Figure 2

L392: Please, can you specify the percentage of piglets born in a cranial position?

Author Response

Thank you for your comments. Please see below a summary of how we have addressed them.

Reviewer 4:

  • I would like to see a paragraph including the different definitions of “eutocia” and “dystocia” in pigs (at least up to now). And afterwards, based on your review, I would expect a more specific conclusion on this topic (further than you stated in L426-L428). Which is your proposal for eutocia/dystocia definition in pigs?

 A new section (2.2. Eutocia and dystocia) has been added and the concluding remarks have been expanded to highlighted areas of further research.

  • Despite high variability between animals, is possible to propose which are the numbers to take under consideration at field level (in terms of duration of farrowing, frequency of manual intervention, piglet survival..)?

 This has been added in the new section”Eutocia and dystocia”

  • L16:The abstract is relatively short. Maybe you can add some main results of the review to arouse the interest of the reader?

 Main results have been added to the abstract

  • L32-L33:I would appreciate a reference supporting that sentence “much research has focused on prevention of problems in ruminants”

  • References have been added

  • L36:Additional information about how “normal” parturition has been documented/defined up to now

  • This is now included in the new section (2. Eutocia and dystocia)
  • L41:Are you referred to hyperprolific sows, or not necessarily? Not necessarily

  • L67:The mention of some papers based on surveys aiming to rank the different painful moments in pigs are welcome. For instance, Ison SH, Rutherford KMD 2014. http://dx.doi.org/10.1016/j.tvjl.2014.10.003 (and probably there are some more papers published more recently).

 Content and references to this and other papers has been added to this section.

  • L99-L112:There is not any reference thought all these text. Please, add.

 References have been added to this section.

  • Section 2.1:A part of figure 1, maybe can help a table summarizing the 3 phases of farrowing, and explaining in each one: the duration, the main hormones, the main clinical effects (contractions, dilatation…)

 Thank you for this suggestion, but we think that this would create a very complex and unwieldy figure and the information is better presented in text.

  • Section 3.1.3:Taking into account that several farms are synchronizing farrowing using PGF2-alpha synthetic, it would be nice to mention it. And reviewing if that management can contribute to increase dystocia, or that depends on the day that is administered…or at least mention that there is another gap related to that (if that is the case).

 Information on this has now been added in the new section (6. Common interventions)

  • L182:(here or wherever) It would be nice to specify the different rhythm and intensity of the myometrial contraction depending on the moment of farrowing.

 This information has been added here

  • L231:It has been reported that the average parturition time is 3.58h for hyperprolific sows…and after how many hours is considered dystocia? After 5 hours for instance?

 This has been added in the new “eutocia and dystocia” section

  • L217:Again, here you mention the dystocia rate, but it would be necessary to know how dystocia (or difficult farrowing) has been defined here.

 This definition has been added in the new section (5. Eutocia and dystocia)

  • L251:Good point, thank you!

 Thank you!

  • L293:Not sure if I understand that sentence… “the secretion of oxytocin behaviours are hindered”..Please, check it.

 Corrected

  • New section 4.3.3:It would be nice to see a paragraph about environmental conditions that can affect farrowing. That should include the optimum temperature/humidity for farrowing, as well as the minimum number of days that sows can enter in the maternity unit in order to acclimate and adapt to the new building.

 A section (4.3.3 Environmental conditions) has been added

  • L336:What is considered high body condition? (introduce the mm of back fat coverage or at least the body condition score)

 This has been added

  • L352(here or wherever): I suggest mention the importance of the administration of enough quantity of water around farrowing. Not sure if there is enough scientific information on this topic, but if not, it can be mention as an important gap to consider.

 This has been added to the 4.5 Sow nutrition section

  • L378:Please, introduce reference of the Figure 2

 This is our own figure

  • L392:Please, can you specify the percentage of piglets born in a cranial position?

 This has been added

Round 2

Reviewer 4 Report

Thank you very much to answer and considere my comments/suggestions. In my opinion, your review can be published in the present form.